# Ontogenetic Variation in Macrocyclic and Hemicyclic Poplar Rust Fungi

**DOI:** 10.3390/ijms232113062

**Published:** 2022-10-27

**Authors:** Zhongdong Yu, Zijia Peng, Mei Qi, Wei Zheng

**Affiliations:** College of Forestry, Northwest A & F University, Xianyang 712100, China

**Keywords:** *Melampsora*, mating type gene, ontogeny, macrocyclic, hemicyclic

## Abstract

*Melampsora larici-populina* (*Mlp*), *M. medusae* (*Mmed*), *M. magnusiana* (*Mmag*), and *M. pruinosae* (*Mpr*) are epidemic rust fungi in China. The first two are macrocyclic rust fungi distributed in temperate humid environments. The latter two are hemicyclic rusts, mainly distributed in arid and semi-arid areas. Ontogenetic variation that comes with this arid-resistance is of great interest—and may help us predict the influence of a warmer, drier, climate on fungal phylogeny. To compare the differences in the life history and ontogeny between the two types of rust, we cloned mating type genes, *STE3.4* and *STE3.3* using RACE-smart technology. Protein structures, functions, and mutant loci were compared across each species. We also used microscopy to compare visible cytological differences at each life stage for the fungal species, looking for variation in structure and developmental timing. Quantitative PCR technology was used to check the expression of nuclear fusion and division genes downstream of *STE3.3* and *STE3.4*. Encoding amino acids of STE3.3 and STE3.4 in hemicyclic rusts are shorter than these in the macrocyclic rusts. Both STE3.3 and STE3.4 interact with a protein kinase superfamily member EGG12818 and an E3 ubiquitin protein ligase EGG09709 directly, and activating G-beta conformational changes. The mutation at site 74th amino acid in the conserved transmembrane domain of STE3.3 ascribes to a positive selection, in which alanine (Ala) is changed to phenylalanine (Phe) in hemicyclic rusts, and a mutation with Tyr lost at site 387th in STE3.4, where it is the binding site for β-D-Glucan. These mutants are speculated corresponding to the insensitivity of hemicyclic rust pheromone receptors to interact with MFa pheromones, and lead to Mnd1 unexpressed in teliospora, and they result in the diploid nuclei division failure and the sexual stage missing in the life cycle. A Phylogenic tree based on *STE3.4* gene suggests these two rust types diverged about 14.36 million years ago. Although these rusts share a similar uredia and telia stage, they show markedly different wintering strategies. Hemicyclic rusts overwinter in the poplar buds endophytically, their urediniospores developing thicker cell walls. They form haustoria with a collar-like extrahaustorial membrane neck and induce host thickened callose cell walls, all ontogenetic adaptations to arid environments.

## 1. Introduction

*Melampsora* have extremely variable life cycles and a correspondingly variable suite of behaviors [1]. Some species are macrocyclic fungi, such as *M**elampsora*
*l**arici-populina* and *M*. *m**edusae*, a quarantine species in China. These species require two different host plants, poplar and larch, to complete a life cycle [2], distributed in humid and cool regions. Asexual reproduction occurs on poplar trees and produces urediniospores and teliospores. Sexual reproduction takes place separately on alternative hosts, e.g., larch needles, and produces pycniospore and acieospores. Other *Melampsora* species are hemicyclic, such as *M*. *m**agnusiana* and *M*. *p**ruinosae*, which are asexual, with shorter, two-stage reproductive cycles even if there are alternative hosts near around [3,4], and distributed in semi-humid or arid regions. We are unaware, however, how this hemicyclic strategy has evolved in these fungi. Little work has explored either its evolution or the mechanisms by which the hemicyclic life strategy is carried out. In short, we are interested in defining exactly what leads to a hemicyclic strategy, and, furthermore, why this hemicyclic life happens.

It is presumed that these changes in life history are ultimately controlled by “mating type” genes [5]. *Melampsora* are basidiomycetes, which have three types of encoding genes involved in mating, Homeodomain (*HD*), Prohormone genes, and G-protein coupled receptors (*GPCRs*) with 7 transmembrane domains, as that of ascomycetes [6]. Homeodomain genes encode HD1 and HD2 homeodomain transcription factors [7,8]. In most basidiomycetes, the *HD1* and *HD2* genes are arranged within the mating type locus as a pair of divergently transcribed genes. For a compatible mating, the product of an *HD1* gene has to interact with the product of an *HD2* gene from a gene pair of another mating type specificity. This specific pairing leads to the formation of a dimer. Dimerization between paired proteins produce an active heterodimeric HD1-HD2 transcription factor that is nuclear localized, binds to specific sequences in the promoters of target genes, and regulates expression of genes acting in dikaryon development and sexual reproduction [9,10]. In this, pheromones and pheromone receptors from mating type loci in *Melampsora* are homologous to similar structures in ascomycetous yeast *Saccharomyces cerevisiae* [11,12]. In *M. larici-populina*, these pheromone receptor genes are usually labelled *STE3* and the pheromone precursor genes are called *Ph* (for pheromone) [13]. The genes encoding pheromone precursors and their receptors (P/R locus) are usually linked at a locus specific to mating type, and in tetrapolar species are unlinked to the homeodomain transcription factor gene locus [14]; however, in bipolar system, the two mating type loci are either linked or fused [15]. Binding of a pheromone to a pheromone receptor usually induces a mating reaction when the ligand and the receptor are products of nuclei of different mating types. Binding of a pheromone stimulates conformational changes at the C-terminus of the pheromone receptor, which then interacts with a heterotrimeric G protein. The newly activated G protein then transmit the pheromone signal via the pheromone-activated signal transduction cascade, e.g., PKA (protein kinase A, PKA) pathway and MAPK (mitogen-activated kinase) pathway [16,17,18]. Regulations evoked by the homeodomain transcription factors is interconnected with this outcome of the activated pheromone signaling cascade [19,20,21], and culminates ultimately in a transcriptional program in the nucleus [22,23,24], and nuclei fusion and division genes (*Kar5*, *Kar9*, *Spo11,* and *Mnd1*) start to express chronologically [25,26].

In this study, the full length of the mating type gene *STE3.3* and *STE3.4* was amplified from four rust species. We compared hemicyclic rusts to macrocyclic rusts, with the intent to study how mutations of mating type gene influenced sexual development of hemicyclic rusts and what happened at each stage of ontogeny in the life history. The amino acid sequence of each of these genes was studied across all four species and the mutation loci and divergence times were analyzed. Sequential expression of nuclear fusion genes (*Kar5*, *Kar9*) and nuclear division genes (*Spo11* and *Mnd1*) in the two rust types were detected by quantitative PCR. At the same time, electron microscopy was used to check for visual structural differences in ontology of the two lifecycle rust types.

## 2. Results

### 2.1. STE3.3 and STE3.4 Structure and Function Prediction

The total length of the *STE3.3* gene ORF in the macrocyclic rusts (Mlp, Mmed) is 1158 bps, encoding 386 amino acids. However, *STE3.3* in Mmag is missing and full length cloning was aborted, the ORF of the *S**TE3.3* gene in hemicyclic rust Mpr has a total length of 1152 bps and encodes 384 amino acids. The ORF of the *S**TE3.4* gene in macrocyclic rusts is 1245 bps, encoding 414 amino acids. The ORF gene *S**TE3.4* in hemicyclic rusts has a total length of 1212 bps and encodes 404 amino acids (Figure 1; Appendix A). The mating type receptors STE3 in hemicyclic rusts are shorter than that of macrocyclic rusts, but with most similar features. For example, both STE3.3 and STE3.4 proteins in both macrocyclic rusts and hemicyclic rusts are alkali, steady, and hydrophobic (Appendix A), with 7 and 6 transmembrane helix, respectively (Appendix A); they both aggregated in the endoplasmic reticulum of fungal cells (Appendix A). Both rusts have two conserved domains, the STE3 and 7tm_GPCRs domains (Figure 2). Both belong to D type family member of G proteins with seven transmembrane domains. The STE3 domain is located between the 12th and 293rd amino acid in STE3.3 protein, 7tm_GPCRs domain at 7th to 295th; while in STE3.4 protein, the STE3 domain was located between the 11th and 293rd, 7tm_GPCRs at 8th to 293rd amino acid site (Figure 2). Their secondary structure of both proteins is composed of mainly α-helixes and random coils, but also a smaller diverse collection of extended strands and beta turns (Appendix A). However, our findings suggest that these proteins did vary slightly in their amino acid composition. STE3.3 protein were composed of Leu8, Tyr9, Leu12, Thr215, Trp270, Val273, Thr274, Ser277, and Thr278, which ultimately bind to signal protein 1U19 protein according to a 4n4wA template in MlpSTE3.3. Conversely, Thr334, Ile336, Tyr387, and Thr391 bind to β-D-glucan in MlpSTE3.4.

Both MlpSTE3.3 and MlpSTE3.4 can interact with protein kinase superfamily (EGG12818) and E3 ubiquitin ligase (EGG09709) directly through prediction by software STRING, and STE3.3 can act with one G-beta protein EGG09074, and a putative DNA-activated nuclear serine/threonine protein kinase EGG08298, Figure 3.

### 2.2. Divergence between Macrocyclic and Hemicyclic Rusts

There was a 96.89% homogenesis between six of the STE3.3 amino acid sequences, but 30 mutant loci and 3 depletions of amino acid, including 12 mutations and the loss of one binding locus (Leu8) were located on the conserved transmembrane domain. A 90.51% homogenesis existed among 8 of STE3.4 amino acid sequences, though there were 73 mutant loci and 12 depletion loci, including 30 mutations at the conserved transmembrane domain and the mutation of 1 binding locus (Tyr387), Figure 1. Molecular phylogenic trees constructed from the *STE3.4* genes show that both macrocyclic and hemicyclic rusts of *Melampsora* were closely grouped in one monophyletic clad. These trees suggest that these fungi diverged about 14.36 million years ago (Figure 4).

The 74th locus at the conserved transmembrane domain of the STE3.3 in hemicyclic rusts is a positive selection site, which mutated from L (in the macrocyclic rusts) to F. The 387th site in the hemicyclic rusts STE3.4 protein was mutated from Y (in the macrocyclic rusts) to S, which is a locus of where proteins are ultimately bounding to β-D-dextran (Appendix A).

### 2.3. Ontogenetic Variation of Urediniospora Stage

Morphologically, Mlp and Mmed urediniospora are broadly ellipsoid or oblong, echinulate with smooth patch occurred apically and equatorially at surface, respectively. Cell wall of Mlp is approximately 2 µm thick, and thickened bilaterally up to 7 µm, while in Mmed the wall is uniformly 2 to 3 µm thick or slight thickened apically (Figure 5(A1,A2)). Mmag and Mpr urediniospores are globose or ovate-globose, and finely verrucose without smooth patches on the surface (Figure 5(A3,A4)), cell wall thickness 2.5 to 4 µm and 7 to 10 µm in the water suspension, respectively.

Development in the macrocyclic and hemicyclic rusts are almost similar. First, germ tubes pass through the stomata to produce a binucleated substomatal sac (Figure 5(B1–B4)), and then develop dikaryotic stalk-haustorium (Figure 5(C1–C4)). However, in Mmag and Mpr, we noted the production of a fold of collar-like extrahaustorial membranes and a thickened host cell wall callosum on either side of the haustoria neck, while Mlp and Mmed never developed these huastorial appendages (Figure 5D1–D4). Third, dikaryotic intercellular hypha accumulate under the epidermis of the host poplars and subsequently develop uredia within nearly 7 days (Figure 5(E1–E4)). These hyphae first differentiate into conidiogenous cells, which usually have two nuclei (Figure 5(F1–F4)). The inner wall of the conidiogenous cell apically protrudes outward to the outer wall of the cell, and the spore sprout is formed first. Subsequently, as the spore sprout extends to a certain length, the apical cells initially form a binuclear urediniospore, before finally developing into urediniospores separated by interval cells.

### 2.4. Ontology in the Telia Stage

The newly produced teliospores were all binucleated (Figure 6A,D,G,J). Both types of rust teliospores were formed by the extension and division of conidiogenous cells, with a high electron density protoplasm and many fat particles inside (Figure 6C,F,I,L). As the teliospores of Mlp developed, the nuclear fusion occurred by day 35 (Figure 6B,E,H,K), and the *Kar9* gene displayed the highest expression at the 30th day and the *Kar5* gene at the 35d (Figure 7A). The nuclear fusion of Mmed teliospores occurred on the 20th to 25th day (Figure 6E), while the *Kar9* gene expressed topically at the 20th day and the *Kar5* gene at the 25th day (Figure 7A). However, the nuclear division genes *Mnd1* and *Spo11* were expressed the highest in Mlp and Mmed at the 40th day (Figure 7A). Most of the macrocyclic rusts overwinter by the fusion teliospore on dropped poplar leaves, and produce mononucleate basidiospore by meiosis in the next spring, usually, the mononucleate basidiospora undergo a mitosis and then get into dikaryotic basidiospore, and initiate infecting larch needles, Appendix A.

Both Mmag and Mpr teliospores were binuclear at the beginning (Figure 6G,J), though fused into a diploid mononuclear (Figure 6H,K) by the 40th day after teliospora maturated. Compared to the macrocyclic rusts above, Mmag and Mpr teliospores were marked by noticeably higher fat content than Mlp and Mmed (Figure 6I,L). The expression of nuclear fusion genes increased significantly in the telia stage of Mmag and Mpr rusts, but failed to express the nuclear division genes *Mnd1* and *Spo11* (Figure 7B). The hemicyclic rusts seem overwinter with dikaryotic teliospora on the dropped leaves. However, telia of hemicyclic rusts, Mmag and Mpr, cannot germinate to produce basidium and basidiospores at all, their roles in lifecycle are not definite.

### 2.5. Ontology of Pycinospora and Aeciospora Stage

No pycnia and aecim were detected in hemicyclic rusts. In the macrocylic rusts, however, pycnia developed on adaxial needles 7 to 10 days after inoculation of Mlp and Mmed basidiospora (Figure 8(A1,A2)). Under a fluorescence microscope, the pycnia of Mlp are flask-shaped (Figure 8(B1,C1)), and the pycniospora are monokaryotic, elliptical, or spherical and have a smooth surface (Figure 8(D1)). Mmed pycnia, however, are hemispherical (Figure 8(B2,C2)), and the pycniospora are mostly elliptical and monokaryotic with a few angular and smooth surfaces (Figure 8(D2)). Aecia developed on the abaxial needle 7 days after pycnia appeared (Figure 8(A3,A4)). Aecium is “U” shaped, with aeciospores strung in tandem (Figure 8(B3,B4)). Mlp aeciospores were spherical and binucleated (Figure 8(C3)), while Mmed rust spores are more elliptical, although also binucleated (Figure 8(C4)). The verrucose on the surface of the Mlp aeciospores are thicker and shorter than those on the surface of the Mmed aeciospores (Figure 8(D3,D4)). Fluorescence microscopy and transmission electron microscopy observed that the aeciosporophora (AP), internal cells (ICs), conidiogenous cells (SP), and aeciospore initiates (AI) arose during aeciospore formation were all dual nuclei (Figure 8(C4,E3,E4)). The cells at the base of the aecia (below the dotted line) were mononuclear (Appendix A) with two monocytes undergoing membrane fusion to become dikaryotic aeciospore cells (Appendix A).

### 2.6. Detection of Rust in Winter Buds of Poplars

PCR of *P. purdomii*, *P. deltoides* cv. “*Zhonghua hongye*” buds indicated that neither of the poplar species was colonized by macrocyclic rust. However, PCR revealed that *P. tomentosa* and *P. euphratica* buds were colonized by rust mycelium. Quantitative PCR indicated hemicyclic rust biomass in *P. tomentosa* winter buds in January 2019 was 6.73 times and 9.35 times that of June 2018 and August 2019, respectively, Figure 7C. In *P. euphratica* buds, q-PCR of pruinosae-ITS showed the biomass of rust in winter buds in January 2019 was 3.55 times and 10.14 times that of June 2018 and August 2019 (Figure 7C). Both *P. tomentosa* and *P. euphratica* winter buds have found intercellular hyphae, with the typically binucleated mycelium (Figure 9E–H), while we did not detect mycelium in Mlp and Mmed winter buds. Artificial inoculated buds were also not developed by any mycelium, and there was no detecting rust biomass in the inoculated winter buds. Hemicyclic rusts can overwinter in poplar buds via mycelium endophytically, and the lifecycle of two types of rust was summarized in Figure 10.

## 3. Discussion

Hemicyclic rusts are evolved from macrocyclic rusts with the loss of sexual reproduction happening over millions of years [27]. Sexual reproduction is controlled by mating-type genes [28], and the recognition and binding of pheromone receptor (STE3.3, STE3.4) is considered a “first step” in the occurrence of sexual reproduction [29,30]. Therefore, the loss of the sexual reproductive stage of hemicyclic rust may be associated with the disabling of these pheromone receptors. In this study, the full length of two pheromone receptor genes was cloned from macrocyclic rusts and hemicyclic rusts using the RACE-smart method. After comparing the amino acid sequences of the two genes, it was found that the mutation loci mainly occurred in the conserved transmembrane domain, for example, the 74th locus amino acid of the hemicyclic rust STE3.3 protein mutated from L to F by positive selection, and similarly, the binding locus at 387th locus of the STE3.4 protein from Y to S. These mutations cause hemicyclic rusts pheromone receptors to be insensitive to mating pheromones and thus, unable to isomerize G-proteins or otherwise initiate downstream cAMP and MAPK pathways. We can see from the phylogenic tree constructed by the gene *STE3.4* that the hemicyclic rusts and the macrocyclic rusts diverged about 14.36 million years ago, and we anticipate that sexual reproduction has been lost since.

Rusts eventually co-evolved with their hosts [31], thus the loss of sexual reproduction in rusts must be associated with a suite of alternative hosts amenable to non-sexual reproduction. Although there are reports of their alternative hosts, namely genus *Corydalis* and genus *Chelidonium* in the USA and Canada [32], we did not find Mmag parasite in these universal plants in China till now. At present, host alternation in Mmag in China is not clear [33], *Corydalis* was thought as an alternative host of Mmag but we found aecia on this plant are of *M. ferrinii* [34,35], artificial inoculation with urediniospora of Mmag in the field failed, and the teliospores of Mmag may not be important in the life cycle, and we took this rust in China as one of the hemicyclic rusts. Savile believed that many species in the genus *Melampsora* could survive in poplar buds via mycelium, wintering until their second year and developing urediniospora to start the primary infection cycle [3]. This study demonstrates this conclusion to be true. However, artificial inoculating poplar buds with urediniospores indoors did not ensure the infection happened: hyphae were neither found after inoculation in buds. These results suggest that primary infection at next year is not possible by aeciospora from alternative hosts in the field, and that most buds are likely infected instead with endophytic hypha. This may be the result of the long-term co-evolution of Mmag and Mpr with their hosts to adapt to the harsh environment outside.

Scanning electron microscopy found that only Mmed rust strains produced the *appressorium* near stomata, which corresponded to what Spiers and Hopcroft reported [36]. After the germtube pass through the leaf cuticle, the haustoria quickly formed and began to uptake nutrients from the host cells. While the morphology of the haustoria is closely related to the host [37], the haustoria produced by Mlp infesting *P. purdomii* was elliptical, while Mlp infecting resistant host *P. deltoides* cv. “*Zhonghua hongye*” was round and spherical. Nevertheless, when Mmag and Mpr infect *P. tomentosa* and *P. euphratica*, their haustoria were all nearly spherical, indicating some plasticity in this structure and design. Spherical haustoria have a larger surface area in contact with host cells, making them easier to absorb nutrients.

Mmag and Mpr showed significant differences in haustoria development when compared to either Mlp or Mmed strains. Namely, the hemicyclic rusts created a collar-like fold of extrahaustorial membranes around the neck and collar-like callosum infection sites, absent in macrocyclic strains. These special structures have the potential to regulate water flow as a sort of “molecular sieve” to increase the survival of the newly infecting hosts [38]. A fold of extrahaustorial membrane probably arose from the sharp confrontation between the host cell wall and the haustoria. Meanwhile, these sealed cellular also provide a safe room for haustoria, and may allow partial symbionts to seek refuge from a harsh external environment. Spiers and Hopcroft also found a pronounced callosum in the infecting site cell walls when Mlp infected resistant *P. nigra* [39], consistent with expectations [40,41,42]. In a compatible combination, once the haustorium is formed, rusts can achieve nutrients from mesophyll cells, and their intercell hyphae grow fast, few days later, their uredia are developed [42]. The four rusts in this study all had similar processes of uredia stage ontogeny, However, cells of hemicyclic rust are often full of an electron-dense protoplasm and a higher proportion of fat particles, as well as a thicker cell wall which can help them adapt to their harsh environments.

Nuclear fusion and meiosis are the most important nuclear behaviors in the telia stage, they are also the boundary of haploid and diploid stage in the lifecycle [13,43]. Hacquard et al. reveals teliospore nuclear fusion occurred at 25 to 39 days, and meiosis occurred soon after nuclear fusion by qRT-PCR and DAPI dying methods with *M. larici-populina* [44]. Our studies, with Mlp and Mmed inoculating *P. purdomii* and *P. deltoides* cv. respectively, gained almost the same results, 30 to 35 days for nuclear fusion and 35 to 40 days for meiosis in Mlp. In Mmed strains, nuclear fusion was observed within 20 to 25 days and meiosis was observed 25 to 30 days later. Both Mmed fusion and nuclear division happened faster in Mmed strains by 10 days. This may be a contributing explanation to why Mmed is more narrowly distributed than Mlp since it was reported in 2019, and is found mainly in north China and a high-altitude region, where the temperature is relatively low [45,46,47]. Earlier nuclear fusion and division may better prepare rusts to enter periods of winter dormancy earlier.

Teliospora of Mmag and Mpr can undergo nuclear fusion, but due to mating gene mutations and absence of *STE3.4* in Mmag, their downstream cAMP signaling cannot form and cannot go on the meiosis process, and the qRT-PCR results demonstrated that the nuclear division gene *Mnd1* is not expressed in these two rusts teliospora, which may be the reason why Mmag and Mpr teliospora do not germinate at all in their lifecycles, and they undergo a different lifecycle from the macrocyclic rusts, Figure 10. However, due to rusts are obligated parasite and uncultured artificially, we currently cannot verify functions of these mutation genes via knockdown and overexpression technologies.

## 4. Materials and Methods

### 4.1. Materials

In this study we used four rusts, *M. larici-populina* (Mlp) and *M. medusae* (Mmed) collected around temperate and humidity regions, *M.*
*magnusiana* (Mmag) and *M. pruinosae* (Mpr) collected around semi-arid China. Hosts, respectively, are *Populus purdomii*, *P. deltoides* cv. “*Zhonghua hongye*”, *P. tomentosa*, *P. euphratica* (Table 1). The Mlp and Mmed ontology analysis was done using indoor inoculations from established collections. Mmag and Mpr ontogeny studies were done by collecting leaves directly from the field in June 2018, January 2019, and August 2019.

### 4.2. STE3.3 and STE3.4 Cloning

Total RNA of four rusts was extracted using the MiniBEST RNA Extraction Kit (Code No. 9769, TaKaRa). Complementary DNA (cDNA) was synthesized using iScript ™cDNA Kit (Code No. RR036A, TaKaRa). According to *STE3.3* (Protein ID 123740) and *STE3.4* (Protein ID 86096) descriptions outlined in the *M. larici-populina* genome-wide database (JGI, http://genome.jgi-psf.org/Mellp1/Mellp1.home.html, accessed on 16 December 2018), we designed primers of *STE3. 3* and *STE3.4* sequences (Table 2) using the software Premier 5, and performed 3′ end and 5′ end RACE amplification, respectively. A high-Fidelity DNA Polymerase, Phanta Super-Fidelity DNA Polymerase (TaKaRa), was employed for intermediate fragment amplification. Products from this PCR amplification were then added by a polyA tail in favor of the following clone and sequencing. RACE primers of the 3′ end and the 5′ end were designed based on the intermediate fragments, and a sequence ATTACGCCAAGCTT was added at the forward of both 3′ end and 5′ end primers to fulfill the seamless clone requirement for full length DNA (Table 2). Vectors for clone were pMD19-T, pUC19, Linearized pRACE vector (Tiangen Ltd., Beijing, China). Sequencing was delegated to Shenggong Ltd. Shanghai, China. The amplified sequences were then aligned together with DNAMAN software to obtain the full-length sequences of the *STE3.4* and *STE3.3* genes, then open reading frame (ORF) was found by software Open Reading Frame Finder (https://www.ncbi.nlm.nih.gov/orffinder/, accessed on 1 November 2020).

### 4.3. Predictions of STE3.3 and STE3.4 Structures and Functions

Nucleotide sequences of *STE3.3* and *STE3.4* were translated into amino acid sequences using the website (https://web.expasy.org/translate/, accessed on 15 January 2021). DNAMAN software was then used to align amino acid sequences and to check mutation loci. GenScript (https://www.genscript.com/psort.html, accessed on 18 January 2021) and Euk-mPLoc 2.0 (http://www.csbio.sjtu.edu.cn/bioinf/euk-multi-2/, accessed on 18 January 2021) were used for predicting subcellular allocation of STE3.3 and STE3.4. Hydrophobicity detection was done using an Expasy protscale online (https://web.expasy.org/protscale/, accessed on 18 January 2021), following methods outlined in reference [48]. Their transmembrane domains were predicted by TMHMM Server v.2.0 (http://www.cbs.dtu.dk/services/TMHMM/, accessed on 18 January 2021), especially for seven-transmembrane segments (TMⅠ-TMⅦ) through analysis of their numbers, loci, and their flanking sequences. We used the SignalP 4.1 Server (http://www.cbs.dtu.dk/services/SignalP-4.1/, accessed on 20 January 2021) to predict signal peptides in STE3.3 & 3.4 using their prebuilt model “Eukaryotes of Organism”, default value of D-cutoff and standard output. We predicted protein secondary structure using SOPMA (https://npsa-prabi.ibcp.fr/cgi-bin/npsa_automat.pl?page=npsasopma.html, accessed on 20 January 2021), and tertiary structure by online software SWISS-MODEL (https://swissmodel.expasy.Org/, accessed on 3 February 2021), and I-TASSER (https://zhang lab.ccmb. med. umich.edu/I-TASSER/, accessed on 3 February 2021). Conserved Domain Database (http://www.ncbi.nlm.nih.gov/Structure/cdd/wrpsb.cgi, accessed on 5 February 2021) was implemented to detect their conserved domains by using database CDD v3.19-58235 PSSM with threshold value 0.01. PAML site model was applied for detecting positively selected loci in STE3.3 and STE3.4, M1 is the null model with omega ≤ 1 for all loci in the nucleotide sequence, while M2 model is positive with any nucleotide locus omega > 1, Chi-square test then was applied for checking the likelihood between the two models with significance value *p* < 1 and freedom of 2, if *p* < 0.05, the null model was denied, and the target genes were recognized as positive selected sites. Online software STRING (https://version11.string-db.org/cgi/input.pl?sessionId=ectguMpuNabQ&input_page_-show_search=on, accessed on 6 March 2021) was employed to predict the interaction web of STE3.3 and STE3.4 by selecting macrocyclic Mlp as the reference organism.

### 4.4. Molecular Clock Phylogenic Tree Construction

*STE3.4* nucleotide sequences of wheat stripe rust (*P. striiformis* f. sp. *tritici*), wheat stalk rust (*P. triticina*), wheat stem rust (*P. graminis* f. sp. *tritici*), and pine west gall Rust (*C. quercuum* f. sp. *fusiforme*) were downloaded from the JGI Database (https://genome.jgi.doe.gov/portal/, accessed on 10 May 2021) to construct the sample molecular clock phylogenic trees. A query of the Timetree database (http://www.timetree.org/, accessed on 10 May 2021) shows that wheat stripe rust and *Melampsora* rust began to differentiate 114 million years ago [49]. Using this calibration point, the differentiation time of the selected sequence was calculated using the MCMCtree software [50]. We calibrated the nodes by using the highest posterior density (HPD) age values for the Melampsora + Puccinia divergence, and used normal distribution prior on the treeModel.rootHeight parameter. The substitution models for STE3.4 was GTR + G + I. Markov chain Monte Carlo (MCMC) was run for 100 million generations, sampling every 1000 generations, to estimate the posterior distribution of clade age and rates. Ten million generations were discarded as burn-in. All effective sample sizes were required to be greater than 200 given that the MCMC has reached convergence (http://tree.bio.ed.ac.uk/software/tracer/, accessed on 10 May 2021). An ultrametric maximum-clade-credibility (MCC) tree was summarized using TreeAnnotator 1.8, including divergence mean times and ranges above 95% probability.

### 4.5. Observation of Ultrastructure at Uredia Stage

Mlp and Mmed urediniospores were inoculated in abaxial surfaces of collected leaves of *P. purdomii* and *P. deltoides* cv “*Zhonghua hongye*”. Seedlings were planted at 25 °C with 16 h light and 8 dark for each day. The 6 h, 1 d, 2 d, 4 d, and 7 d leaves after inoculation were cut into small pieces of 0.5 cm × 0.5 cm for observations under scanning electron microscopy (SEM) and of 0.2 cm × 0.4 cm for observation under transmission electron microscopy (TEM). The preparation methods of SEM and TEM follow those designed in Yao et al. [51]. Mmag and Mpr samplings were collected in the field and were directly prepared for electron microscopy observation as the above.

### 4.6. Ultrastructural Observation and qRT-PCR Analysis at the Telia Stage

Seedlings were inoculated for 10 d before being transferred to a 4 °C chamber. Sampling began every 5 days since first observation of telia. Telia samples of Mmag and Mpr were directly collected in the field. RNA extraction and cDNA synthesis were implemented with SsoFast™EvaGreen Kit (TaKaRa Cat # 172-5201). Nuclear fusion gene and nuclear division gene primers (Table 3) were designed for qRT-PCR according to Hacquard et al. [44]. qRT-PCR reaction conditions referred to Ye et al. [52], and were repeated three times for each reaction. Mlp’s alpha-tubulin (aTUB) and elongation factor-1-alpha (ELF1a) were selected as endogenous reference genes [53]. Gene expression was calculated using the 2^−^^ΔΔCt^ method [44]. Nuclei in the telia stage were detected using fluorescent microcopy (Leica DM4000B) by diamidino-phenyl-indole (DAPI) dying.

### 4.7. Rust Detection and Verification by Artificial Inoculation of Poplar Winter Buds

Each harvested poplar bud was rinsed three times with sterilized water and divided into three parts (Appendix A). One part was used for DNA extraction, one part was used for RNA extraction, and the other part was used for TEM preparation. A pair of specific primers for basidiomycetes fungi [54], ITS_1__F_/ITS_4B_ (1F-CTTGGTCATTTAGAGGGAAGTAA; 4B-CAGGAGAC TTGCACGGTCCAG), were used for PCR with the positive control DNA of each rust urediniospore and the ddH_2_O as the negative control. RNA extraction, cDNA synthesis, and qRT-PCR reactions all followed the methods laid out in Section 4.2. The Mlp-ITS gene primer (Table 3) in qPCR was designed with reference to Boyle et al. [55]. Mlp-ITS was prioritized as the target gene, and a TUB and an ELF1a were included as internal reference genes used to indicate biomass at the sampling time in *P*. *t**omentosa* and *P*. *e**uphratica* species. Artificial inoculation was used to verify whether rust could infect winter buds annually by spraying 1 × 10^8^/mL urediniospora suspensions of Mlp and Mmag on buds of the potted *P*. *p**urdomii* and *P*. *t**omentosa*, respectively. On the 10th day after inoculation, samples were taken and the buds were checked for rust fungi infection with TEM.

### 4.8. Observation of the Ontogeny of the Basidiospores Stage

After one month of the growing telia on the leaves of *P. purdomii* and *P. deltoides* cv. “*Zhonghua hongye*”, the vitro leaves were brought indoors to break teliospore dormancy using freezing/melting and wetting/drying alternating at −20 °C [56]. Then, to induce germination, we kept the leaves under 18 °C and 100% humidity conditions for 1 to 2 days. Leaves with Mmag and Mpr teliospora were collected from the field and treated as the above for checking basidiospores. Nuclei were detected under fluorescence microcopy (Leica DM4000B) by DAPI dying.

### 4.9. Ultrastructural Observation of Pycniospore Stage and Aeciospore Stage

After rusts in *P. purdomii* and *P.*
*d**eltoides* cv. “*Zhonghua*
*h**ongye*” leaves developed basidiospores, the vitro leaves were suspended above needles of the 3-year-old larch seedlings, with the leaves adaxial surface down so that the spores may drop freely on the pine needles. The whole pine seedling was covered with a transparent plastic bag and cultured under 20 °C with photoperiod 16/8 h conditions each day. When pycnia appeared on the adaxial needles, we immediately sampled needles for pycniosprora and pycnia using fluorescence microscopy and TEM.

## 5. Conclusions

Mutations of the mating gene STE3.3 and STE3.4 acceptors in the hemicyclic rusts on poplar trees lead to the loss of sexual reproduction. The nuclear division gene *Mnd1* is unexpressed in hemicyclic teliospores, which in turn leads to the inability of germination, and consequently a lack of pycniospores or aeciospores production. To continue its life cycle, the hemicyclic rust endophytically overwinters in the buds with mycelium before beginning the infection process at the next year, during which it forms a thick urediniospore wall. The development of urediniospores and teliospores is roughly similar in the ontology of the two types of rust, but the hemicyclic rusts developed haustoria with larger surface area and collar-like extrahaustorial membrane neck and induced host cell produce thickened callosum wall, increasing their ability to adapt to the environment.

## Figures and Tables

**Figure 1 ijms-23-13062-f001:**
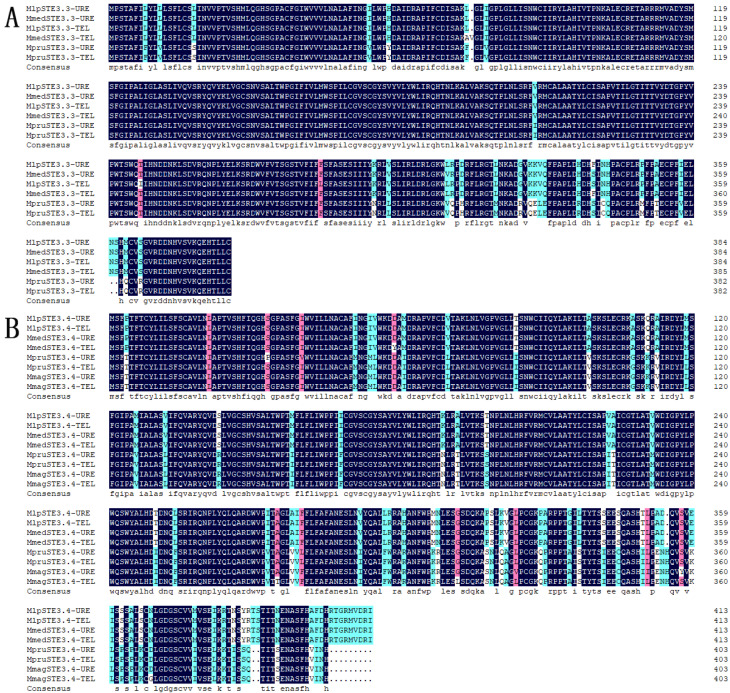
Homologous alignment of amino acid sequences of STE3.3 and STE3.4 in *Melampsora.* Note: Dark background refers to 100% identical; pink background refers to 75% to 100% identical; blue background refers to 50% to 75% identical; white background refers to lower than 30% identical; (**A**), STE3.3; (**B**), STE3.4.

**Figure 2 ijms-23-13062-f002:**
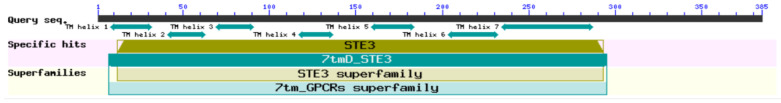
CDD predicts the conserved domain of STE3.3 and STE3.4 protein in Mlp.

**Figure 3 ijms-23-13062-f003:**
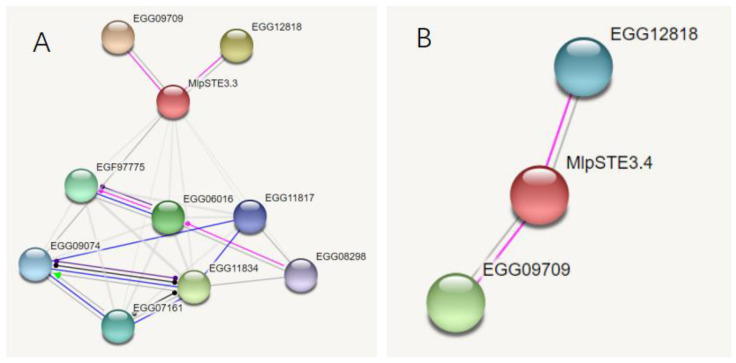
Functional prediction of STE3.3 and STE3.4 protein. Pink lines indicate posttranslational modification, dark blue lines indicate binding, and black lines indicate reaction. (**A**) STE3.3; (**B**) STE3.4.

**Figure 4 ijms-23-13062-f004:**
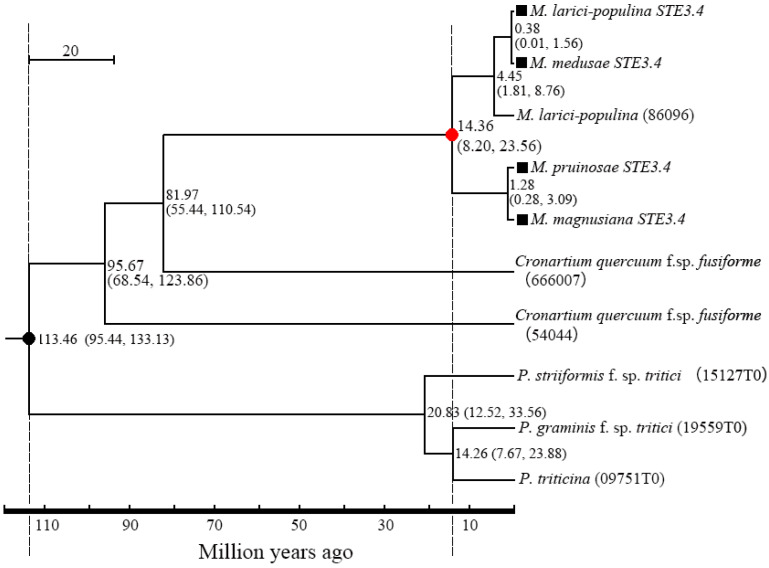
Divergence time between macrocyclic rusts and hemicyclic rusts. Dark dot “●” refers to the calibrate time based on the divergence time between *P. striiformis* f. sp. *tritici* and M. larici-populina; Red dot “●” refers to the divergence time of the tested rusts in this study; ■: the tested rusts; parentheses refers to 95% confidence for the pair’s diverged time.

**Figure 5 ijms-23-13062-f005:**
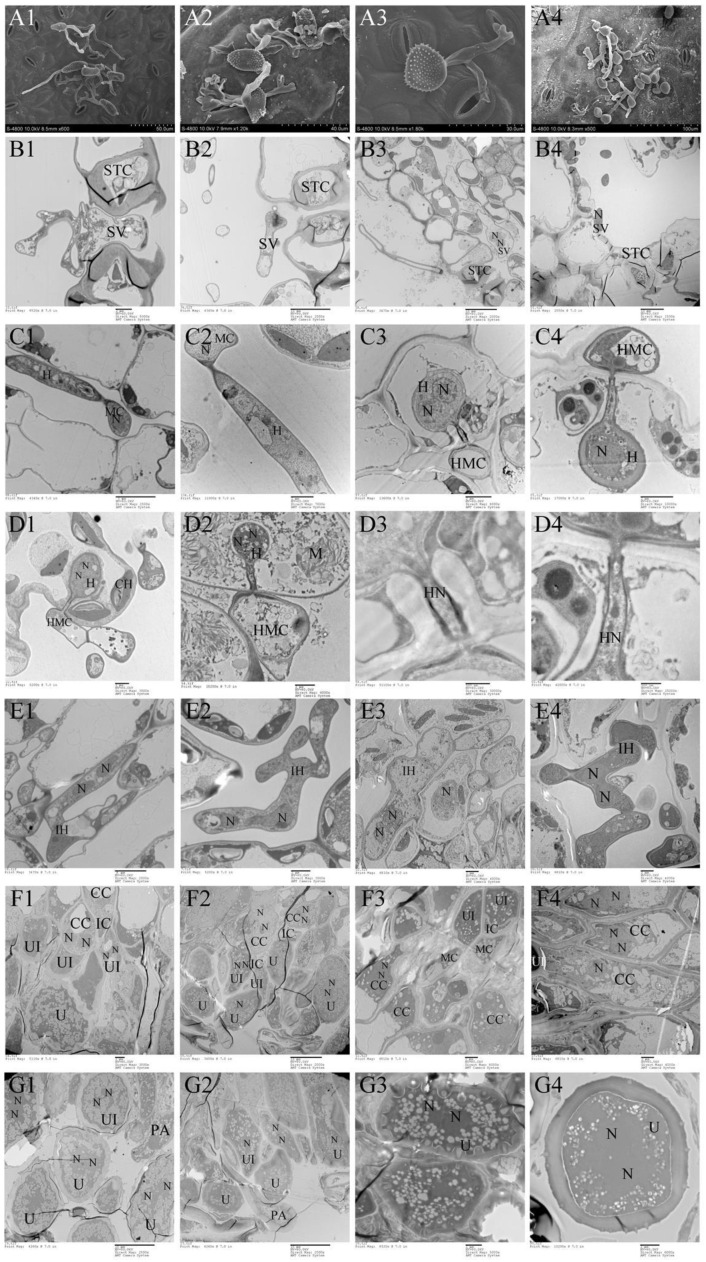
Ontogeny of uredia stage. (**A1**–**G1**), *Mlp*; (**A2**–**G2**), *Mmed*; (**A3**–**G3**), *Mmag*; (**A4**–**G4**), *Mpr*; Appressorium (A); stomatal cell (STC); substomatal vesicle (SV); nuclei (N), mother cell (MC); hypha (H); intercellular hypha (IH); interval cell (IC); haustorium mother cell (HMC); haustorium (H); chloroplast (CH); mitochondria (M); conidiogenous cell (CC); urediniospore initial (UI); urediniospore (U); interval ce;ll (IC); paraphysis (PA).

**Figure 6 ijms-23-13062-f006:**
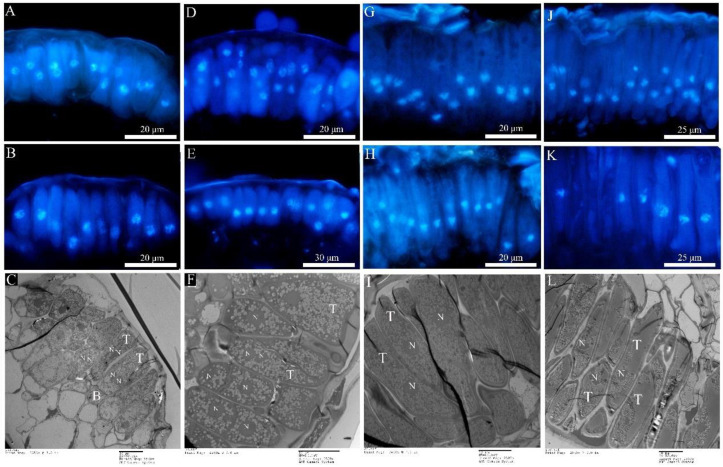
Teliospora under fluorescence microcopy and transmission electron microscopy. (**A**–**C**): *Mlp*; (**D**–**F**): *Mmed*; (**G**–**I**): *Mmag*; (**J**–**L**): *Mpr*; (**A**,**D**,**G**,**J**) refer to dikaryotic teliospore; (**B**,**E**,**H**,**K**) refer to the fusion and mononucleate (diploid) teliospore; (**C**,**F**,**I**,**L**) refer to teliospore under TEM; teliospore (T); basel cell (B); nuclei (N).

**Figure 7 ijms-23-13062-f007:**
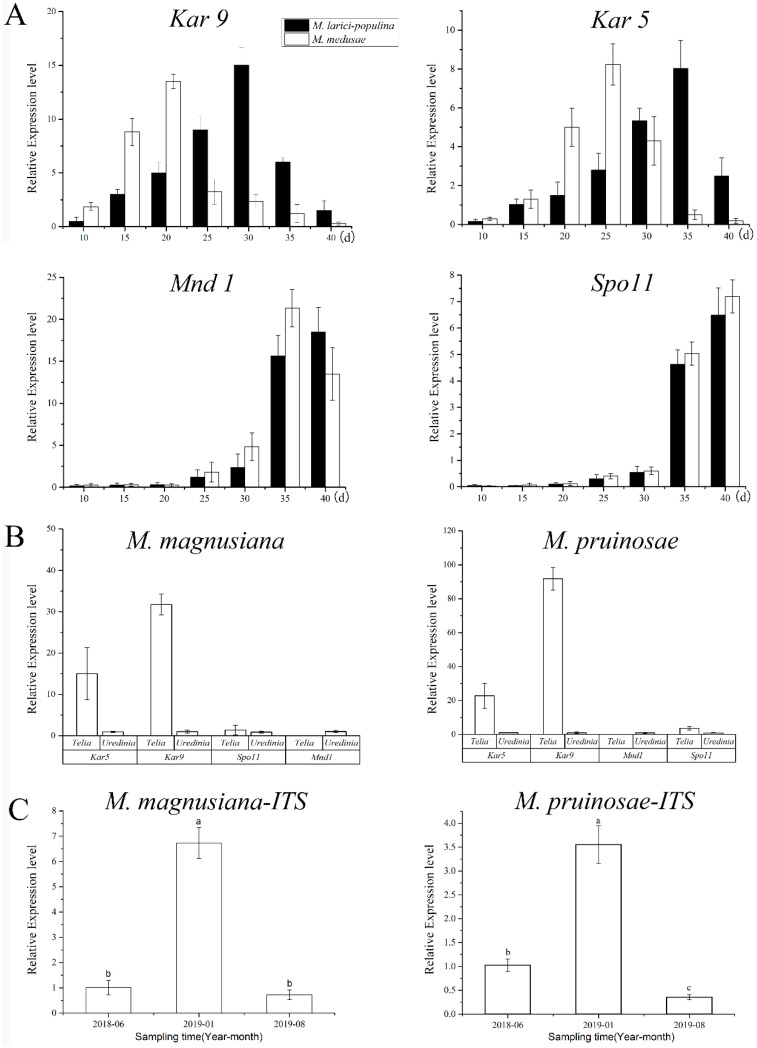
Quantitative PCR detection based on nuclei fusion gene, nuclei division genes, and internal transcript spacer (ITS). Note: (**A**), expressions of nuclei fusion gene (*Kar9*, *Kar5*) and division gens (*Mnd1*, *Spo11*), respectively in MLP telia and in Mmd telia; (**B**), expressions of *Kar9*, *Kar5*, *Mnd1* and *Spo11*, respectively in MLP telia and in Mmd telia; (**C**), Fungal biomass at different collecting times, a–c in the bar charts refer to significant differences at 0.05 level.

**Figure 8 ijms-23-13062-f008:**
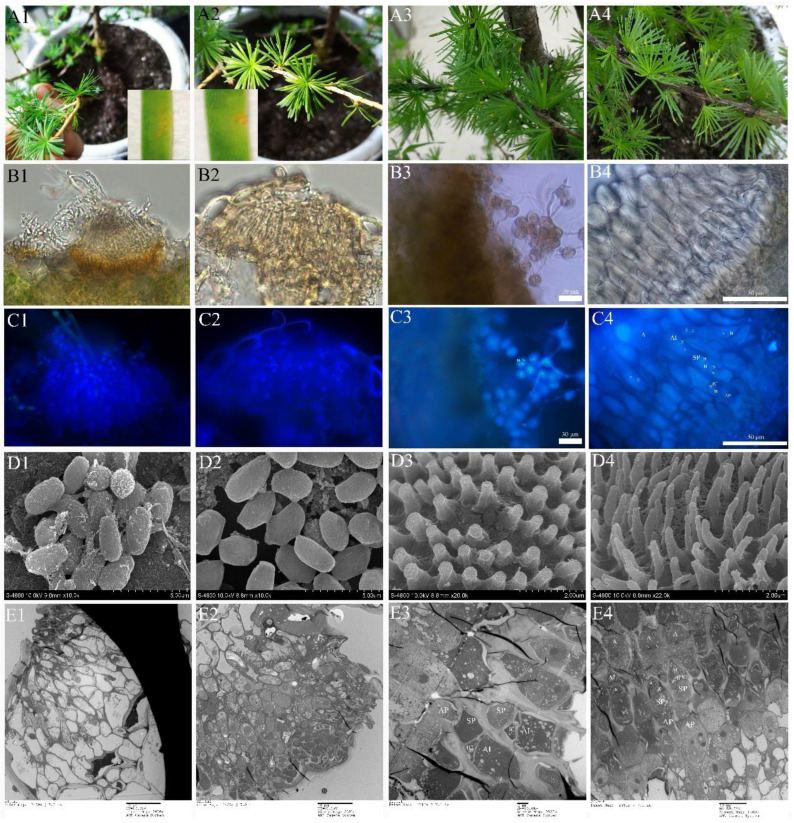
Ontogeny of pycniospora and aeciospora. (**A1**–**E1**), pycinospora stage of Mlp; (**A2**–**E2**), Pycinospora stage of Mmed; (**A3**–**E3**), Aeciospora stage of Mlp; (**A4**–**E4**), Aeciospora stage of Mmed; (**A1**,**A2**), Pycnia on the abaxial needle; (**B1**,**C1**,**B2**,**C2**), Pycnia under fluorescence microcopy; (**D1**,**D2**), Pycniospora under SEM; (**E1**,**E2**), Pycnia under TEM; (**A3**,**A4**), Aecium on the adaxial needle; (**B3**,**C3**) and (**B4**,**C4**) are both aecium under light and fluorescence microcopy; (**D3**,**D4**) Verrucose on the surface of aeciospora under SEM; (**E3**,**E4**), aecium under TEM; pycniospora (S), nucleus (N); conidiogenous cells (SP), interval cell (IC); aeciospora initial (AI); aeciosporophora (AP); aeciospore (A).

**Figure 9 ijms-23-13062-f009:**
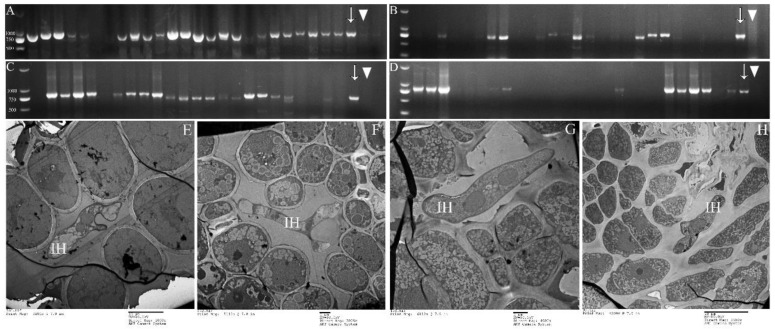
Mycelium detection in buds of *P. tomentosa* and *P. euphratica*. (**A**,**B**): PCR detection for *P. tomentosa* buds; (**C**,**D**): PCR detection for *P. euphratica*; (**E**,**F**): TEM detection for *P. tomentosa buds*; (**G**,**H**): TEM detection for *P. euphratica* buds; (**A**,**C**): sampling in January 2019; (**B**,**D**): sampling in August 2019; Arrow (↓) refers to positive control; triangle (▼) refers to negative control; Intercellular hypha (IH).

**Figure 10 ijms-23-13062-f010:**
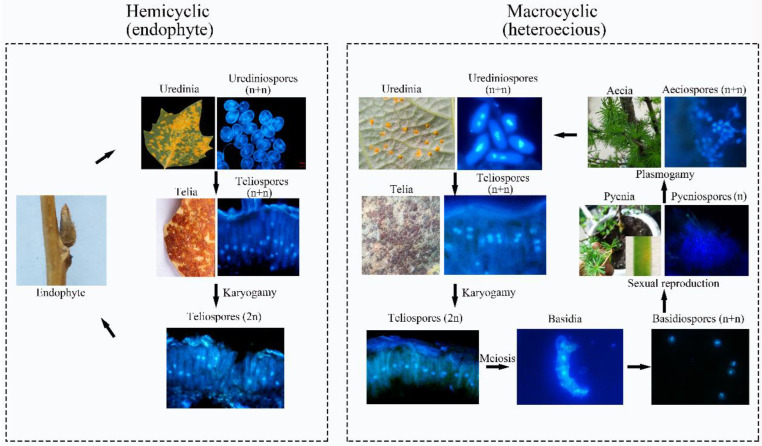
Lifecycle model of the macrocyclic rust and the hemicyclic rust.

**Table 1 ijms-23-13062-t001:** Four samples of *Melampsora* in this study, with host plants, sampling locations, voucher, and GenBank accession numbers.

Host Plants	Locality	Voucher Specimen No.	GenBank Accession		Species
			ITS	D_1_/D_2_	
*P. purdomii*	Taibai Shaanxi	HMAS 247968	MK028576	MK064523	*M. larici-populina*
*P. tomentosa*	Tianshui Gansu	HMAS 247981	MK028582	MK064530	*M. magnusiana*
*P. euphratica*	Inner Mongolia	HMAS 247982	MK028585	MK064533	*M. pruinosae*
*P. deltoides* cv. “*Zhonghua hongye*”	Linyou Shaanxi	HMAS 247973	MK028586	MK064528	*M. medusae*

Note: Samples with a prefix “HMAS” are saved in Mycological Herbarium of Institute of Microbiology, Chinese Academy of Sciences.

**Table 2 ijms-23-13062-t002:** Primer sequences for RACE amplification of *STE3* genes.

Gene	Primer Name	Primer Sequence (5′–3′)	Annealing Temperature
*STE3.3*	5′RACE Outer1	GATTACGCCAAGCTTGTGTTCTGAGGTTCGTGTGCTGGCG	65 °C
	5′RACE Inner1	GATTACGCCAAGCTTAGCCAATACAAC ACATACGCGGAGTA	65 °C
	3′RACE Outer1	GATTACGCCAAGCTTGCTCAA TCTCGT GGGTCCAGTGGGTCTC	69 °C
	3′RACE Inner1	GATTACGCCAAGCTTTGTATCAACTTC AAGCCCGCGACTGG	60 °C
*STE3.4*	5′RACE Outer2	GATTACGCCAAGCTTGCAGCTAGGACG CACATTCGGACA	66 °C
	5′RACE Inner2	GATTACGCCAAGCTTAGCGGTGCAAAT TCAAAGGATTGGTCGA	62 °C
	3′RACE Outer2	GATTACGCCAAGCTTGTCCCTCAATGT GTACCAGGCTC	66 °C
	3′RACE Inner2	GATTACGCCAAGCTTCAATGAACTTGG AA TCCGGGAGTG	58 °C

**Table 3 ijms-23-13062-t003:** qRT-PCR primer sequences for nuclear behavior genes and biomass of rusts.

Gene Symbol	Primer Sequence (5′–3′) Forward/Reverse	Amplicon Length (bp)	Annealing Temperature
*Kar5*	F-GCTAATATCTCTAGCTTCATGCC R-CTTGTAGACCCGGAAACCT	166	53
*Kar9*	F-CATTGTCCCGTTAGCTGGTT R-CTGCTGAAGGTCCACCAAGT	221	55
*Mnd1*	F-AGAACTTGATAGATGATGGATTAG R-GCTTTATTAATGTTCACCGATTG	100	50
*Spo11*	F-AGTGGTTTGTTACTTGGTATTACT R-CATGCTGAGCCGGTAAGT	139	53
*ITS*	F-TGAGCGACTTTAATGTGACTC R-ATGTAAATCAAAGTTGCCTTTGCG	123	55
Alpha-tubulin (*aTUB*)	F-ATCTGTAACGAACCTCCTGCTA R-CCTCCTCCATACCTTCTCCAA	168	55
Elongation factor-1-alpha (*ELF1a*)	F-CGAGACTCCCAAATACTTCGTT R-GTTCACGAGTTTGACCATCCTT	167	55

## Data Availability

The data presented in this study are available in the article and Appendix A. The RNA-seq data used in this study is available in NCBI.

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
