# Peer review of "Ontogenetic Variation in Macrocyclic and Hemicyclic Poplar Rust Fungi"

_ijms, 2022, doi:10.3390/ijms232113062_

Round 1

Reviewer 1 Report

This study is novel and important. The manuscript is well written and can be accepted. 

Author Response

Thank you for your review and supports.

Reviewer 2 Report

In this paper, the author analyzed mating type genes, STE3.4 and STE3.3 to reveal ontogenetic variation of typical polar rust fungi with different life cycle. However, the rust species selected by the author have some problems.

We have doubts about the biological characteristics of two species, making the results appear suspect. The presence of M. medusae in China is the first issue. The abstract for "M. medusae..epidemic rust fungus in China" was written by the author. But that is wholly incorrect. This Melampsora species does not exist in China, yet Chinese Custom continues to treat it as a quarantine fungus, according to Tian & Kakishima (2005). Therefore, M. medusae is not the rust fungus on P. deltoides. If the author is unable to prove the species' identity, its life cycle is likewise suspicious. As a result, none of the results are trustworthy. Prior to conducting any further study or analyzing any data, the author should first confirm the research goal.

In this research, M. magnusiana was referred to as hemicyclic rust; however, it actually has aecial hosts on Chelidonium majus or Corydalis species and telial hosts on polar section Leuce, including Populus davidiana, P. hopeiensis, and others. The rust fungus is macrocyclic.

The main text and references both have numerous errors.

Author Response

Thanks for your suggestion and hard works. Based on your review, we revised the introduction and discussion sections throughout, methods and references are also replenished in the main manuscript and in reference section. The main corners are answered as below:

In this paper, the author analyzed mating type genes, STE3.4 and STE3.3 to reveal ontogenetic variation of typical polar rust fungi with different life cycle. However, the rust species selected by the author have some problems.

We have doubts about the biological characteristics of two species, making the results appear suspect. The presence of M. medusae in China is the first issue. The abstract for "M. medusae...epidemic rust fungus in China" was written by the author. But that is wholly incorrect. This Melampsora species does not exist in China, yet Chinese Custom continues to treat it as a quarantine fungus, according to Tian & Kakishima (2005). Therefore, M. medusae is not the rust fungus on P. deltoides. If the author is unable to prove the species' identity, its life cycle is likewise suspicious. As a result, none of the results are trustworthy. Prior to conducting any further study or analyzing any data, the author should first confirm the research goal.

Answer: Actually, no report of M. medusae was found in China before 2005, and M. medusae has been a quarantine fungus in China till now. It was firstly reported in China by Zheng et al. (2019), and also reported by other researchers as one of eight Melampsora species infected Populus in China after that (Zhou et al. 2020; Jiang et al. 2021). P. deltoides is one known host of M. medusae (Gortari et al. 2018; Zheng et al. 2019; Ji et al.2020), and now numerous poplar clones are distributed across the north China (Cao & Conner 1999). In this year Chinese government carried out investigation of the invasion species in most provinces in China, and M. medusae is listed as one of them.

Please refer to:

Cao, F.L.; Conner, W.H. Selection of flood-tolerant Populus deltoides clones for reforestation projects in China. For. Ecol. Manag. 1999, 117, 211-220. https://doi.org/10.1016/S0378-1127(98)00465-4

Gortari, F.; Guiamet, J.J.; Graciano, C. Plant-pathogen interactions: leaf physiology alterations in poplars infected with rust (Melampsora medusae). Tree physiol. 2018, 38, 925–935.  https://doi.org/10.1093/treephys/tpx174

Jiang, N.; Liang, Y.M.; Tian, C.M. Eight Melampsora species infected in poplars in China. Journal of Northwest Forestry University 2021, 36, 149-157. (in Chinese) https://doi.org/10.3969/j.issn.1001-7461.2021.02.22

Zheng, W.; Newcombe, G.; Hu, D.; Cao, Z.M.; Yu, Z.D.; Peng, Z.J. The first record of a north American poplar leaf rust Fungus, Melampsora medusae, in China. Forests 2019, 10, 182. https://doi.org/10.3390/f10020 182.

Zhou, Y.; Dai, M.L.; Dai, X.G.; Li, S.X.; Yin, T.M.; Li, XP. Isolation and Primary Identification of Pathogen of Leaf Rust on Black Cottonwood (Populus deltoides) in the South of China. Molecular Plant Breeding, 2020, 18, 8240-8246. (in Chinese) https://doi.org/10.13271/j.mpb.018.008240

Ji, J.X.; Li, Z.; Li, Y.; Kakishima, M. Life cycle and taxonomy of Melampsora abietis-populi in China and its phylogenetic position in Melampsora on Populus. Mycological Progress 2020, 19:1281–1291. https://doi.org/10.1007/s11557-020-01624-1

In this research, M. magnusiana was referred to as hemicyclic rust; however, it actually has aecial hosts on Chelidonium majus or Corydalis species and telial hosts on polar section Leuce, including Populus davidiana, P. hopeiensis, and others. The rust fungus is macrocyclic.

Answer: As Tian (2005) reported “In Japan, Hiratsuka (1927, 1932) established that M. magnusiana alternates between Populus sieboldii and Chelidonium majus. According to the records, spermogonia and aecia of M. magnusiana have been found on Corydalis in Hebei province, China (Tai, 1979). This record is doubtful as Miura (1928) reported a rust species, Caeoma fumariae on Corydalis spp. in Caohekou of Liaoning Province, but he did not establish the relationship between M. magnusiana and Caeoma fumariae. M. yezoensis Miyebe et Matsumoto also forms aecia on Corydalis. At present, host alternation in M. magnusiana and M. rostrupii in China is not clear. Field observations and results from inoculation experiments have shown that M. magnusiana overwinters on poplar as uredinial mycelium. Thus, it can survive and propagate itself in many areas where its alternate host is absent.” In fact, Corydalis and Chelidonium are rarely infected and the teliospores of M. magnusiana may not be important in the life cycle. In this study, we found both STE3.3 deletion and mutants in STE3.4 ascribe to the loss of 0 & I stage in the lifecycle of M. magusiana. In China, Corydalis was never found 0 and I stage of M. magnusana since Tai (1979) but of M. ferrinii (Peng et al., 2022). Compared to M. pruinosae, their lifecycles are similar, and we believe M. magnusana is one of hemicyclic rust in China, too. We aimed to compare the divergence between macrocyclic rust and hemicyclic rust on the ontogeny and mating genes, so as to find a clue of two types of rust fungi’ evolution and adaptation when facing to the long and changeable environments. We added the correspondent contents in the introduction and discussion sections, and references are renewed.

Please refer to:

Hiratsuka, N. Inoculation experiments with some heteroecious species of the Melampsoraceae in Japan. Japanese Journal of Botany 1932, 6, 1-33.

Peng, Z.J.; Xiong, C.W.; Luo, Z.Y.; Hu, X.Y.; Yu, Z.D.; Chen, T.X.; Xu, Y.; Wang, B. First report of alternate hosts of willow rust disease caused by Melampsora ferrinii in China. Plant Dis. 2022, 1, 324. https://doi.org/doi:10.1094/PDIS-05-21-0958-PDN.

Tai, F.L. Sylloge Fungorum Sinicorum. Science Press: Beijing, China, 1979; pp 537-539. (in Chinese)

Tian, C.M.; Kakishima, M. In Rust Diseases of Willow and Poplar; Pei, M.H.; McCracken, A.R., Ed.; CABI publishing: London, UK, 2005; Chapter 8, pp 99-112.

The main text and references both have numerous errors.

Answer: we revised the whole manuscript intensively again.

Reviewer 3 Report

Manuscript entitled “Ontogenetic variation in macrocyclic and hemicyclic poplar rust fungi” mainly introduced the evolutionary relationship among four epidemic rust fungi. However, several points need to be addressed before it can be published.

   The most concerned is the selection of mating type genes. There are several important genes were involved in mating, which was introduced in the Introduction section. The author selected two mating type genes, STE3.4 and STE3.3 to analyze the evolutionary relationship among four epidemic rust fungi. So, why the author chose two STE genes, but not a STE gene, together with another type of genes, such as Ph, which was more convincing to reflect the evolutionary relationship by using two different types of mating genes.

    The author compared the differences in the life history and ontogeny between the two types of rust and expected to predict of the influence of a warmer, drier, climate on fungal phylogeny. However, only if the four epidemic rust fungi were originated from a common ancestor, it had significance to compare the evolutionary difference among four epidemic rust fungi from different geographical locations with distinct environment.

 Minor points:

1. Table 2, primers for STE 3.4 was absent.

2. Figure 7, significance difference needed to be added in Figure 7A and B.

3. 2.4 Detailed parameters of constructing molecular clock phylogenic tree need to be added.

Author Response

Manuscript entitled “Ontogenetic variation in macrocyclic and hemicyclic poplar rust fungi” mainly introduced the evolutionary relationship among four epidemic rust fungi. However, several points need to be addressed before it can be published.

   The most concerned is the selection of mating type genes. There are several important genes were involved in mating, which was introduced in the Introduction section. The author selected two mating type genes, STE3.4 and STE3.3 to analyze the evolutionary relationship among four epidemic rust fungi. So, why the author chose two STE genes, but not a STE gene, together with another type of genes, such as Ph, which was more convincing to reflect the evolutionary relationship by using two different types of mating genes.

Answer: Very good suggestion. There are many sets of mating type genes, e.g. STE1, STE2, STE3, and another type of Ph. We searched throughout references and found STE3.3 and STE3.4 are two mating type genes of being intensively studies and easy for us referring. In these four rust fungi, we attempted to clone more mating genes, also inclusive of Ph, but we can’t be always successful. In spite of this, we still fail to achieve the full length of STE3.3 in M. magnusana.

The author compared the differences in the life history and ontogeny between the two types of rust and expected to predict of the influence of a warmer, drier, climate on fungal phylogeny. However, only if the four epidemic rust fungi were originated from a common ancestor, it had significance to compare the evolutionary difference among four epidemic rust fungi from different geographical locations with distinct environment.

Answer: Yes, there are about 20 species of Melampsora reported in China currently, most of them are heteroecious life cycle and distributed across China sporadically. Only these four rusts are found prevalent in China at present, and are distributed in distinctive regions; therefore, we conceived an idea that why they are different in lifecycle and distribution.

 Minor points:

  1. Table 2, primers for STE 3.4 was absent.

Answer: we revised already.

  1. Figure 7, significance difference needed to be added in Figure 7A and B.

Answer: thanks a lot, we added additional notes for Figure 7A and B.

  1. 2.4 Detailed parameters of constructing molecular clock phylogenic tree need to be added

Answer: we added detailed parameters for constructing the tree.

Round 2

Reviewer 2 Report

One last comments, confirm your "Melampsora medusae"  is the authentic one.  I have emphasized that this Melampsora species does not exist in China, yet Chinese Custom continues to treat it as a quarantine fungus, according to Tian & Kakishima (2005). But the author did not revise related part.

Author Response

One last comments, confirm your "Melampsora medusae"  is the authentic one.  I have emphasized that this Melampsora species does not exist in China, yet Chinese Custom continues to treat it as a quarantine fungus, according to Tian & Kakishima (2005). But the author did not revise related part.

Answer: We appreciate you for your comments and suggestion. We confirm M. medusae is an authentic species prevailed in China. We revised the related part in this version and addressed it is a quarantine species in China in the section Introduction, and discussed its distribution definitely in the section Discussion. We also added papers of Tian CM himself (2021) and Zhou et al.(2020) as our references .

In fact, Tian & Kakishima (2005) did not talk about M. medusae but express as “Thus, we consider that the previously reported M. abietis-canadensis (Farl.) Ludwig in China is a consequence of misidentification. Melampsora abietis-canadensis is regarded as a synonym of M. medusae (Cellerino, 1999)”, In 2021, Tian himself and other coauthor reported and described M. medusae as one of eight prevalent rust fungi in China (Jiang et al.,2021). Another researcher also reported M. medusae occurred in south China (Zhou et al.,2020).

References:

  1. Jiang, N.; Liang, Y.M.; Tian, C.M. Eight Melampsora species infected in poplars in China. Journal of Northwest Forestry University 2021, 36, 149-157. (in Chinese) https://doi.org/10.3969/j.issn.1001-7461.2021.02.22
  1. Zheng, W.; Newcombe, G.; Hu, D.; Cao, Z.M.; Yu, Z.D.; Peng, Z.J. The first record of a north American poplar leaf rust Fungus, Melampsora medusae, in China. Forests 2019, 10, 182.https://doi.org/10.3390/f10020 182.
  1. Zhou, Y.; Dai, M.L.; Dai, X.G.; Li, S.X.; Yin, T.M.; Li, XP. Isolation and Primary Identification of Pathogen of Leaf Rust on Black Cottonwood (Populus deltoides) in the South of China. Molecular PlantBreeding, 2020, 18, 8240-8246. (in Chinese) https://doi.org/10.13271/j.mpb.018.008240

Reviewer 3 Report

All the comments were addressed.

Author Response

Thank you for your hard works and kindness.